# Serum 25-Hydroxy Vitamin D Levels in Children with Acute Respiratory Infections Caused by Respiratory Virus or Atypical Pathogen Infection

**DOI:** 10.3390/nu15061486

**Published:** 2023-03-20

**Authors:** Lu Kuang, Zhuofu Liang, Changbing Wang, Tao Lin, Yingying Zhang, Bing Zhu

**Affiliations:** 1Center Laboratory, Guangzhou Women and Children’s Medical Center, Guangzhou Medical University, Guangzhou 510120, China; kuanglu@gwcmc.org (L.K.);; 2Clinical Laboratory, Guangzhou Women and Children’s Medical Center, Guangzhou Medical University, Guangzhou 510120, China

**Keywords:** vitamin D, acute lower respiratory infections, viral infection, intensive care, pediatric

## Abstract

We aimed to clarify the involvement of vitamin D status in virus or atypical pathogens infection in children with acute respiratory infections (ARIs). In this retrospective study, 295 patients with ARIs were attacked by a respiratory virus or a single atypical pathogen; 17 patients with ARIs induced by two pathogens, and 636 healthy children were included. Serum 25(OH)D levels of all children were measured. Oropharyngeal samples of the patients for viruses or atypical pathogens were studied by polymerase chain reaction (PCR) or reverse transcription-polymerase chain reaction (RT-PCR). In our studies, 58.98% of the 295 single-infected subjects and 76.47% of the 17 co-infected subjects had 25(OH)D levels below the recommended 50.0 nmol/L; the mean 25(OH)D levels were 48.48 ± 19.91 nmol/L and 44.12 ± 12.78 nmol/L. Low serum 25(OH)D levels were remarkable in patients with one of seven viruses or atypical pathogens infected. These results were significantly different from those in the healthy group. There were no significant differences in 25(OH)D levels between single infection and co-infection groups. There were no differences in severity among means of 25(OH)D levels. Female or >6-year-old children patients with low serum 25(OH)D levels were more vulnerable to pathogenic respiratory pathogens. However, serum 25(OH)D levels may be related to the recovery of ARIs. These findings provide additional evidence for the development of strategies to prevent ARIs in children.

## 1. Introduction

Acute respiratory infections (ARIs) are an important cause of acute illnesses and mortality worldwide and in China [1,2,3,4]. Respiratory virus or atypical pathogens are the major cause of ARIs. The predominant viruses that caused ARIs in children are respiratory syncytial virus and the influenza virus [5,6,7,8]. Vitamin D is well known for its important role in bone and calcium stabilization [9,10]. Serum 25(OH)D levels are measured to determine vitamin D status, as it is the main circulating form of vitamin D [11]. Recently, an increasing number of studies have indicated that low levels of vitamin D may play an important role in the occurrence and development of extraskeletal diseases because of its immunoregulation and anti-inflammatory effects [10,12,13,14]: vitamin D modulated pathogen recognition receptor expression; increased antimicrobial peptide expression; modulated autophagosome production of inflammatory cytokines reducing apoptosis; and modulated production and chemokine expression [15,16,17,18]. In situations where vaccines do not prevent most ARIs diseases caused by viruses [19], the use of vitamin D as an intervention is likely to be important in reducing ARIs. In addition, the association between vitamin D status and different respiratory pathogens has been poorly studied. To clarify the involvement of vitamin D status in virus- or atypical pathogens-induced infection in children with ARIs, we analyzed 25(OH)D levels and evaluated the prevalence of vitamin D deficiency in them. Controlling ARIs and chronic lung disease progression is likely to benefit from this intervention.

## 2. Materials and Methods

This retrospective study included a total of 948 children aged 0–16 years from Guangzhou Women and Children’s Medical Center between January 2020 and December 2023. Among them, two hundred and ninety-five hospitalized children with ARIs were singly infected by a virus or an atypical pathogen; seventeen patients with ARIs were induced by two pathogens infection. Other 636 healthy children of similar gender and age, who attended the Child Health Clinic, were taken as the control group.

Exclusion criteria: (1) infected by bacteria, fungi, or parasitic; (2) suffering from malnutrition; (3) immune deficiency and liver and kidney insufficiency; (4) vitamin D supplements; (5) treated with glucocorticoid and other immune modulators; (6) abnormal serum phosphorus or calcium level; and (7) hepatic and renal enzymes in the serum that indicate an abnormality.

Control subjects were children admitted during the same period without respiratory infections or any of the exclusion criteria.

The serum samples for 25(OH)D quantitative detection in this study were measured by means of 25-Hydroxy Vitamin D ELA KIT (Immunodiagnostic Systems Limited, Boldon, UK) following manufacturing instruction, and the assay used the biotin-avidin system enzyme-linked immune (BAS-ELISA) method. An oropharyngeal swab for polymerase chain reaction (PCR) and reverse transcription-polymerase chain reaction (RT-PCR) for respiratory viruses and atypical pathogens was collected. Each sample was simultaneously tested for human adenovirus (HAdVs), human enterovirus (EV), parainfluenza virus (PIV), human metapneumovirus (hMPV), severe acute respiratory syndrome coronavirus 2 (SARS-CoV-2), respiratory syncytial virus (RSV), *Mycoplasma pneumonia* (MP), influenza A virus (FA), and influenza B virus (FB).

We defined vitamin D status categories based on the 25(OH)D cut-off levels [20,21] as follows: a serum 25(OH)D level of 75.1–150.0 nmol/L is indicative of vitamin D sufficiency (VDS) status; a level of 50.1–75.0 nmol/L as vitamin D insufficiency (VDI); a level of 25.1–50.0 nmol/L as Vitamin D deficiency (VDD); and a level of ≤25.0 nmol/L as severe vitamin D deficiency (SVDD).

Statistical analysis was performed using SPSS 19.0 and Graph Pad Prism 8. To compare vitamin D levels between groups, independent samples *t*-tests and one-way analysis of variance were used. To compare vitamin D levels in patients before and after hospitalization, paired sample *t*-test was used. Vitamin D status was compared among groups using the Chi-square test. Logistic regression analysis was used for predicting risk factors for those infected. For all tests, a *p* value < 0.05 was considered as significant.

Informed consent was obtained from the patients’ legal guardians, and the study was approved by the hospital-based ethics committee. (Ethic Committee Name: Ethics Committee of Guangzhou Women and Children Medical Center; Approval Code: 202043001; Approval Date: 13 July 2020).

## 3. Results

### 3.1. Characteristics of the Study Cohort

Among the 636 Child Health Clinic children, whose ages were between 1 month and 15 years old, the median age was 72 months (interquartile range of 25th–75th percentiles (IQR): 36–108 months), and 377 (59.28%) were male. Among the 295 hospitalized children with ARIs attacked by a single virus or/and atypical pathogen, whose ages were between 1 month and 16 years old, the median age was 72 months (interquartile range (IQR) 24–108 months), and 158 (53.56%) were male (Table 1). In addition, 17 patients had been co-infected with two pathogens (Table 2), and 5 (29.41%) patients had co-infections, with the majority due to HAdVs + EV-induced infection. As can be seen, the healthy group presented superior serum 25(OH)D levels to ARIs (*t* = 9.84, *p* < 0.0001) or the co-infections group (*t* = 3.783, *p* = 0.0002), but there was no difference in gender and age (*p* > 0.05). In the ARIs group, we compared the mean serum 25(OH)D levels between the ARIs group and the co-infection group (48.48 ± 19.91 vs. 44.12 ± 12.78 nmol/L) and found no significant difference in them (*t* = 0.93, *p* = 0.35).

### 3.2. The Prevalence of VDD and SVDD in Children with ARIs Was Significantly Higher Than That in Healthy Children

As shown in Table 1, of the 295 children with ARIs, only 9.83% (*n* = 29) subjects had VDS, 30.51% (*n* = 90) had VDI, 50.51% (*n* = 149) had VDD, and 9.15% (*n* = 27) had SVDD; in the healthy groups, there were 26.42% (*n* = 168) subjects in VDS status, 42.45% (*n* = 270) had VDI, 30.19% (*n* = 192) had VDD, and only 0.94% (*n* = 6) had SVDD; none were found to have serum 25(OH)D levels above 150 nmol/L. It is also shown that there is a higher percentage of VDS and VDI populations in the healthy group than in the ARIs group. In contrast to that, the ARIs group has higher rates of VDD and SVDD than the healthy group. There are significant differences in the vitamin D status between the healthy and ARIs groups based on our analysis (*X*^2^ = 94.66, *p* < 0.0001). Therefore, we suggest that children who have serum 25(OH)D levels below 50 nmol/L (VDD and SVDD status) are at potential severe risk of ARIs.

### 3.3. Compared with Healthy Children, EV, and FB-Infected Children Did Not Have Significantly Low Serum 25(OH)D Levels

In the 295 children with ARIs, there were 65 (22.03%) subjects with HAdVs infection, 49 (16.61%) with PIV, 44 (14.92%) with EV, 37 (12.54%) with hMPV, 36 (12.20%) with SARS-CoV-2, 24 (8.14%) with RSV, 21 (7.12%) with MP, 9 (3.05%) with FA, and 10 (3.39%) with FB (Figure 1a). Figure 1b presents serum 25(OH)D mean values distributed among the ARIs group, co-infection group, and healthy group. The ARIs group (48.48 ± 19.91 nmol/L) showed statistically significantly (*t* = 10.28, *p* < 0.0001) lower level of serum 25(OH)D compared to the healthy group (63.37 ± 21.15 nmol/L). In addition, we analyzed the serum 25(OH)D levels of the different pathogen infection groups with the healthy group and found that low serum 25(OH)D levels showed no statistical significance in the EV (*t* = 1.84, *p* = 0.073) or FB group (*t* = 1.49, *p* = 0.17) compared with the healthy group (Figure 1c).

### 3.4. Serum 25(OH)D Levels of ARIs Children Admitted to the Pediatric Ward and Intensive Care Unit Were Not Significantly Different

Among the 295 hospitalized children, there were 54 (18.31%) children admitted to the intensive care unit (ICU). As shown in Table 3, the serum 25(OH)D levels of children in the pediatric ward and ICU were 48.44 ± 19.39 and 43.64 ± 23.79 nmol/L (*t* = 1.574, *p* = 0.17). In addition, serum 25(OH)D levels of patients with different respiratory pathogens infection were not significantly different between patients admitted to the pediatric ward and ICU (*p* > 0.05).

### 3.5. Female or >6-Year-Old Children with Low Serum 25(OH)D Levels Were More Vulnerable to Pathogenic Respiratory Pathogens Than Male Children or <6-Year Age Groups

As shown in Table 4, female children with VDD and SVDD were more vulnerable to a virus or an atypical pathogen than males (*p* = 0.002). In addition, the prevalence of VDD and SVDD was significantly different (*p* = 0.0006) between patients < 6 and >6 years old. Eleven (45.83%) out of twenty-four patients infected with SARS-CoV-2, and four (44.44%) out of nine infected with FA, were younger than one year of age. Patients with VDD or SVDD had a higher infection rate of EV, RSV, PIV, hMPV, or MP in the >6 years age group than in other age groups. Results of logistic regression analysis for predicting risk factors of ARIs are given in Table 5. The results showed that female children (OR = 1.329, 95%CI: 1.007–1.755, *p* = 0.045), >6-year-old (OR = 2.587, 95%CI: 1.946–3.438, *p* < 0.0001), and with a vitamin D status of VDD and SVDD (OR = 3.264, 95%CI: 2.450–4.349, *p* < 0.0001) was the risk ratio for ARIs, which was statistically significant.

### 3.6. Serum 25(OH)D Levels Rise in Children with ARIs before Discharge from Hospitalization

Serum 25(OH)D levels were measured in 15 children with ARIs attacked by a virus or an atypical pathogen at hospital admission and at discharge after recovery. Eleven out of fifteen children had higher serum 25(OH)D levels after recovering from ARIs than before. As shown in Figure 2, the children that recovered from ARIs had significantly higher (*p* = 0.0208) serum 25(OH)D levels than when they were admitted.

## 4. Discussion

Vitamin D is mainly converted from 7-dehydrocholesterol via Ultraviolet B (UVB) radiation on skin, and a small amount of vitamin D can also be obtained from the diet. Vitamin D is metabolized by the liver into 25(OH)D and then by the kidney into 1,25-dihydroxyvitamin D. It is estimated that >85% of serum 25(OH)D is used by local target tissues for autocrine/paracrine activation to calcitriol [22]. The 25(OH)D level in serum is frequently used to assess vitamin D status since it’s the major form of vitamin D in circulation [23,24]. Globally, around 1 billion people are estimated to be deficient or inadequate in vitamin D [25]. In China, vitamin D deficiency (<50 nmol/L) was found in 21.3% of males and 43.6% of females [26]. Insufficient 25(OH)D may be a risk factor for 25-hydroxyvitamin D levels in patients with chronic disease [22]. In our study, 174 out of 295 single-infected and 13 out of 17 co-infected patients showed serum 25(OH)D levels lower than 50 nmol/L (VDD and SVDD). The main physiological function of vitamin D is to maintain the balance of calcium and phosphorus metabolism and promote bone health [10]. Previous studies [27,28,29] have pointed out that vitamin D deficiency is associated with many chronic diseases (autoimmune disease, cardiovascular disease, diabetes, obesity, mental disease, malignant tumor, etc.) including osteoporosis, caries, rickets, and other vitamin D deficiency diseases in children. However, none of the subjects in our study had these conditions. We suggest that children with ARIs induced by viruses or atypical pathogens infection present lower serum 25(OH)D levels than healthy children.

There are numerous immune-modulatory functions that vitamin D exhibits in vitro. Vitamin D induces antimicrobial peptides (cathelicidin and defensines) and regulates Toll-like receptors (TLR); vitamin D also induces the development of dendritic cells and regulatory T cells, and, moreover, it regulates the inflammatory cascade by modulating the nuclear factor-kB pathway [15,16]. These modulatory functions rely on an adequate level of 25 (OH) D in the cycle, and an insufficient level can lead to abnormal regulation. In this study, we found the prevalence of VDD and SVDD in children with ARIs was significantly higher than that in healthy children (59.66% vs. 31.13%). Additionally, there is also a higher proportion of healthy children in VDS and VDI groups compared to ARIs; the *p* value has shown that the difference was less significant for VDS and VDI (*X*^2^ = 8.008, *p* = 0.005) rather than for VDD and SVDD in ARIs groups. In agreement with a previous study [30], we suggest that a serum 25(OH)D concentration of 50 nmol/L or higher, which is believed to be the normal level, has been shown to reduce the risk of developing ARIs in children.

Some researchers have reported on the effects of vitamin D on the respiratory syncytial virus [31,32], influenza virus [33], severe acute respiratory syndrome coronavirus 2 [34,35], and rhinovirus infection [36]. However, the main viral etiological agents responsible for ARIs are not limited to those mentioned earlier [2,4]. We investigated serum 25(OH) D levels in hospitalized children with different respiratory pathogens or atypical pathogen infections. Our results indicate that low 25(OH)D levels are obvious in patients who were infected with HAdVs, PIV, hMPV, SARS-CoV-2, RSV, MP, or FA (Table 1; Figure 1). EV- and FB-infected children did not have significantly low serum 25(OH)D levels in the ARIs group, which may be due to the fact that we excluded the subjects infected by EV with non-respiratory complications (myocarditis, meningoencephalitis, myelitis, etc. [37,38]), and to the small sample size of patients with FB. Our results showed that vitamin D levels may decrease in children with various infections. However, statistical results have shown an association but not a causal relationship. These limitations failed to provide significant results to support our conclusion. In order to determine whether vitamin D deficiency is responsible for reduced infections or responsible for lifestyles, cross-sectional studies are urgently needed.

As in the previous study [39], vitamin D supplementation reduces proinflammatory cytokines in the lung and promotes viral clearance by modulating macrophage and T lymphocyte activity. A researcher studied the effect of vitamin D administration on the outcome of patients with Ventilator-Associated Pneumonia (VAP) who had a high rate of mortality. The results have indicated that vitamin D administration can significantly reduce the IL-6 levels as prognostic marker for VAP patients [40]. In COVID-19, higher vitamin D levels correlate with lower interleukin 6 levels, which are important targets for controlling cytokine storms [41]. In our study, vitamin D levels rose when symptoms were relieved. As we showed (Figure 2), vitamin D levels rose after recovery from 15 children with ARIs. However, the exact relationship between vitamin D levels and viral clearance has not been studied. It is not clear whether elevated vitamin D levels are a result of recovery or a cause of recovery. The influence of vitamin levels on disease or the effects of disease on vitamin levels may be modified by characteristics of a child’s environment, background, and diet. Additionally, differences in sunlight-dependent production and circulation of vitamin D are influenced by age [20]. Due to the small number of patients, more comprehensive studies are needed to reach a consensus on this issue.

A lower serum 25(OH)D level was found in patients in the ICU than in the pediatric ward, but this difference was not statistically significant (Table 3). This may be due to the small sample size of ICU patients. According to these results, in our opinion, vitamin D deficiency affects the susceptibility to respiratory viruses and atypical pathogens without affecting the severity of the infection. In addition, female patients over 6 years of age and with serum 25(OH)D levels below 50 nmol/L were more susceptible to respiratory viral infections than male patients (Table 4). Vitamin D deficiency has been reported as an increased risk of acute lower respiratory infections in children [39]. In our study, we found that being female, being >6 years old, or having vitamin D levels < 50 nmol/L may be risk factors for children with ARIs (Table 5). We suggest that children older than 6 years of age or who are female may be at greater risk of acute respiratory infections due to less exposure to sunlight during outdoor activities. This result is slightly different from previous reports that preschool girls with low vitamin D levels are more likely to be infected with the respiratory virus [42]; it may be attributed to the difference in geographical location and lifestyles in different countries.

In vitro, the immunoregulatory activities of vitamin D have been reported [43,44], which affects both the innate as well as adaptive immune responses. IL-6 production in response to RSV stimulation was reduced by vitamin D. Influenza virus infection is reduced in a mouse model by vitamin D through suppressing inflammatory cytokines and reducing virus replication; it is also reduced by the inhibiting effect of vitamin D on platelet activation. Rhinovirus replication and release were inhibited by vitamin D through cathelicidin induction. However, the exact role of vitamin D in the pathogenesis of ARIs has not been determined, and it may be involved in the development and control of viruses through a variety of mechanisms [15,16,17,18]. Most of the ARIs diseases caused by pathogens are not vaccine-preventable [45,46]. Considering this, the immunomodulatory properties of vitamin D may be crucial to controlling both ARIs and the immunopathology associated with lung disease development. Vitamin D supplements have been shown to be used in the treatment of infectious diseases [31,32,33,34,35,36].

The present study’s limitations include (1) a lack of detailed diet information; (2) a small sample of patients who suffered severe attacks; and (3) a lack of data about vitamin D supplements for children with ARIs. In order to confirm whether vitamin D supplements can reverse the unsatisfactory results associated with vitamin D deficiency in ARIs subjects, randomized interventions are needed.

In summary, in this retrospective study, we found that low serum 25(OH)D levels were associated with a respiratory virus or atypical pathogen infection, and not with the severity of ARIs in children. Being female, being >6 years old, or having vitamin D levels < 50 nmol/L may be risk factors for children with ARIs. The immunomodulatory effect of vitamin D should arouse our high attention. We encourage a continued assessment of serum 25(OH)D levels in children with ARIs.

## Figures and Tables

**Figure 1 nutrients-15-01486-f001:**
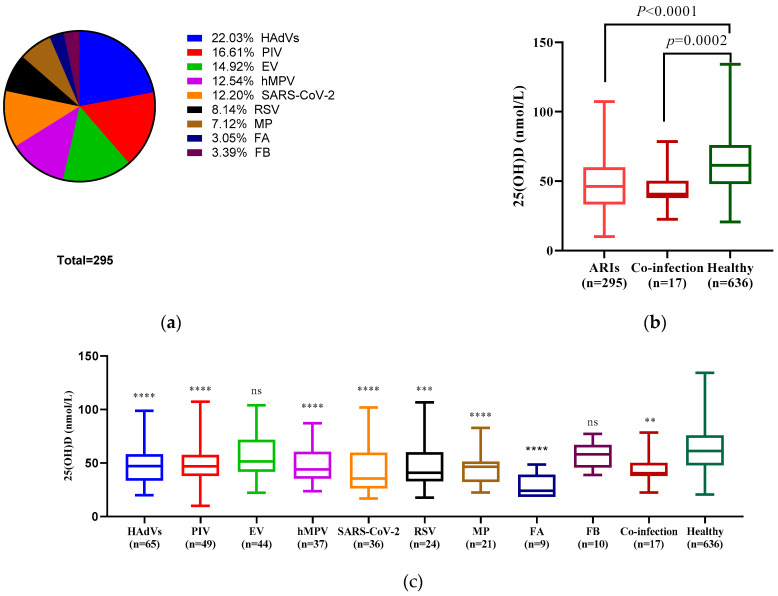
Pathogens distribution and serum 25(OH)D levels comparison. (**a**) Pie diagrams depicting the distribution of the 295 ARIs children hospitalized caused by 9 pathogens. (**b**) Serum 25(OH)D levels in ARIs, co-infection, and healthy group. There was a statistically significant difference between the ARIs or co-infection and healthy groups (*p* < 0.0001, *p* = 0.0002). (**c**) Serum 25(OH)D levels in hospitalized children with ARIs caused by different pathogens and the healthy group. Statistically significant differences were found between the seven groups and the healthy group (****: *p* < 0.0001, ***: *p* < 0.0005, **: *p* = 0.0058). The boxes in the graphs (**b**,**c**) indicate the first quartile and third quartile, and the bars indicate the maximum, second quartile, and minimum values. Statistical analysis was performed using the unpaired *t*-test and one-way analysis. (Abbreviations: ARIs, acute respiratory infections; HAdVs, human adenoviruses; EV, human enterovirus; PIV, parainfluenza virus; hMPV, human metapneumovirus; SARS-CoV-2, severe acute respiratory syndrome coronavirus 2; RSV, respiratory syncytial virus; MP, *Mycoplasma pneumonia*; FA, influenza A virus; FB, influenza B virus).

**Figure 2 nutrients-15-01486-f002:**
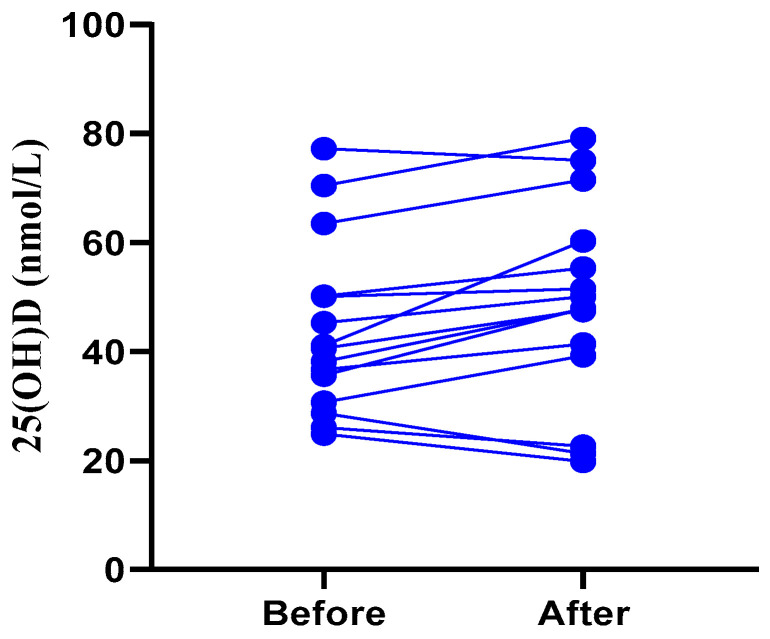
The comparison of serum 25(OH)D levels in 15 ARIs children before and after hospitalization. Statistical analysis was performed using the paired samples *t*-test, *p* = 0.0208.

**Table 1 nutrients-15-01486-t001:** Characteristics and vitamin D status between children hospitalized with ARIs and healthy children.

Characteristic	ARIs (*n* = 295)	Co-Infection (*n* = 17)	Healthy (*n* = 636)	*p* Value
Age (months)	72 (24–108)	52 (2–120)	72 (36–108)	>0.5 ^1^
Gender (Male:Female)	158:137	10:7	377:259	>0.5 ^2^
25(OH)D levels (nmol/L)	48.48 ± 19.91	44.12 ± 12.78	63.37 ± 21.15	<0.0001 ^3^/=0.0002 ^4^
Vitamin D Status VDS (75.1–150.0 nmol/L) VDI (50.1–75.0 nmol/L) VDD (25.1–50.0 nmol/L) SVDD (≤25.0 nmol/L)				
	29 (9.83)	1 (5.88)	168 (26.42)	<0.0001 ^5^
	90 (30.51)	3 (17.65)	270 (42.45)	
	149 (50.51)	11 (64.71)	192 (30.19)	
	27 (9.15)	2 (11.76)	6 (0.94)	

Abbreviations: ARIs, acute respiratory infections; VDS, vitamin D sufficiency; VDI, vitamin D insufficiency; VDD, vitamin D deficiency; SVDD, severe vitamin D deficiency. Ages are presented as median (interquartile range of 25th–75th percentiles); serum 25(OH)D levels are reported as mean ± SD; vitamin D statuses are reported as *n* (%); ^1^
*p* value of ARIs vs. healthy according to age; ^2^
*p* value of ARIs vs. healthy according to gender; ^3^
*p* value of ARIs vs. healthy according to serum 25(OH)D levels; ^4^
*p* value of co-infection vs. healthy according to serum 25(OH)D levels; ^5^
*p* value of ARIs vs. healthy according to vitamin D status.

**Table 2 nutrients-15-01486-t002:** The distribution of co-infected pathogens in children with ARIs (*n* = 17).

Pathogens	Patients, *n* (%)
HAdVs + EV	5 (29.41)
HAdVs + PIV	2 (11.76)
HAdVs + MP	2 (11.76)
EV + PIV	2 (11.76)
RSV + PIV	2 (11.76)
RSV + hMPV	2 (11.76)
hMPV + MP	1 (5.88)
PIV + MP	1 (5.88)

Abbreviations: HAdVs, human adenoviruses; EV, human enterovirus; PIV, parainfluenza virus; hMPV, human metapneumovirus; RSV, respiratory syncytial virus; MP, *Mycoplasma pneumonia*. Reported as *n* (%).

**Table 3 nutrients-15-01486-t003:** The 25(OH)D levels in children hospitalized with ARIs between the pediatric ward and ICU (*n* = 295).

Pathogens	Pediatric Ward *n* = 241 (%)	Admission to ICU *n* = 54 (%)	*p* Value
HAdVs	48.18 ± 17.96 (21.16)	44.42 ± 25.69 (24.07)	0.64
EV	56.02 ± 21.25 (16.18)	47.60 ± 34.69 (9.26)	0.87
PIV	49.07 ± 18.83 (17.01)	44.53 ± 26.77 (14.81)	0.61
hMPV	48.60 ± 16.83 (13.69)	40.82 ± 21.74 (9.26)	0.47
SARS-CoV-2	43.84 ± 25.89 (10.79)	45.09 ± 26.79 (16.67)	0.83
RSV	44.32 ± 15.84 (7.88)	48.20 ± 23.07 (7.41)	0.85
MP	45.00 ± 14.87 (7.05)	37.54 ± 17.39 (9.26)	0.48
FA	26.48 ± 11.91 (2.49)	38.92 ± 17.84 (9.26)	0.75
FB	56.80 ± 13.12 (3.73)	-	-
Total	48.44 ± 19.39 (99.98)	43.64 ± 23.79 (100.00)	0.12

Abbreviations: ARIs, acute respiratory infections; ICU, intensive care unit; HAdVs, human adenoviruses; EV, human enterovirus; PIV, parainfluenza virus; hMPV, human metapneumovirus; SARS-CoV-2, severe acute respiratory syndrome coronavirus 2; RSV, respiratory syncytial virus; MP, *Mycoplasma pneumonia*; FA, influenza A virus; FB, influenza B virus. Reported as mean ± SD nmol/L (*n*%). “-”: there were no ICU admissions due to FB.

**Table 4 nutrients-15-01486-t004:** Numbers of children infected by pathogens in 174 VDD + SVDD subjects according to gender and ages.

Pathogens	Patients	Gender	Ages			
		Female	0–1 y	1–3 y	3–6 y	>6 y
HAdVs	40	24 (60.00)	13 (32.50)	1 (2.50)	4 (10.00)	22 (55.00)
EV	20	12 (60.00)	6 (30.00)	0 (-)	5 (25.00)	9 (45.00)
PIV	27	15 (55.56)	5 (18.52)	0 (-)	3 (11.11)	19 (70.37)
HMPV	22	16 (72.73)	2 (9.09)	1 (4.55)	5 (22.73)	14 (63.64)
SARS-CoV-2	24	8 (33.33)	11 (45.83)	4 (16.67)	4 (16.67)	5 (20.83)
RSV	14	13 (92.86)	1 (7.14)	0 (-)	1 (7.14)	12 (85.71)
MP	15	10 (66.67)	5 (33.33)	0 (-)	2 (13.33)	8 (53.33)
FA	9	2 (22.22)	4 (44.44)	4 (44.44)	0 (-)	1 (11.11)
FB	3	0 (-)	0 (-)	0 (-)	2 (66.67)	1 (33.33)
Total	174	100 (57.47) *	47 (27.01)	10 (5.75)	26 (14.94)	91 (52.30) **

Abbreviations: VDD, 25(OH)D level 25.1–50.0 nmol/L; SVDD,25(OH)D level ≤ 25.0 nmol/L; y, years; HAdVs, human adenoviruses; EV, human enterovirus; PIV, parainfluenza virus; hMPV, human metapneumovirus; SARS-CoV-2, severe acute respiratory syndrome coronavirus 2; RSV, respiratory syncytial virus; MP, *Mycoplasma pneumonia*; FA, influenza A virus; FB, influenza B virus; y, years. Reported as *n* (%). “-”: not available; *: male vs. female, *p* = 0.002; **: <6 y vs. >6 y, *p* = 0.0006.

**Table 5 nutrients-15-01486-t005:** Results of logistic regression analysis with ARIs vs. healthy group.

	OR	95% CI for OR	*p* Value
Gender (Female)	1.329	1.007–1.755	0.045
Age (>6-year-old) VDD + SVDD	2.587	1.946–3.438	0.000 0.000
	3.264	2.450–4.349	

Abbreviations: CI, confidence interval; OR, odds ratio; VDD, 25(OH)D level 25.1–50.0 nmol/L; SVDD,25(OH)D level ≤ 25.0 nmol/L.

## Data Availability

Not applicable.

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
