# Peer review of "Serum 25-Hydroxy Vitamin D Levels in Children with Acute Respiratory Infections Caused by Respiratory Virus or Atypical Pathogen Infection"

_nutrients, 2023, doi:10.3390/nu15061486_

Round 1
Reviewer 1 Report
The article studied and compared the serum Vitamin D Levels in Children with Acute Respiratory Infections. It is a well done article. However, I still have 2 comments. 1. Line 187: The activation pass way of vitamin D had been discussed in the article “New Vitamin D analogs and changing therapeutic paradigms” reported in Kidney International (2011) 79, 702–707; doi:10.1038/ki.2010.387. Vitamin D activation occurred 85% in local tissue that related to health of local organ and less than 15% of vitamin D activation in kidney that related to Calcium absorption.  Please put some discussion in this point of view. 2. Line 195: Previous studies have pointed out that vitamin D deficiency is associated with many chronic diseases (autoimmune disease, cardiovascular disease, diabetes, obesity, mental disease and malignant tumor, etc.) except osteoporosis, caries, rickets and other vitamin D deficiency diseases in children……  “Except” is wrong. It should be ‘including”.
Author Response
Point 1: Line 187: The activation pass way of vitamin D had been discussed in the article “New Vitamin D analogs and changing therapeutic paradigms” reported in Kidney International (2011) 79, 702–707; doi:10.1038/ki.2010.387. Vitamin D activation occurred 85% in local tissue that related to health of local organ and less than 15% of vitamin D activation in kidney that related to Calcium absorption.  Please put some discussion in this point of view.
Response 1: We thank the reviewer for the constructive comment. We have added this in the discussion section.". It is estimated that >85% of serum 25(OH)D is used by local target tissues for autocrine/paracrine activation to calcitriol [22]. ... Insufficient 25(OH)D may be a risk factor for 25-hydroxyvitamin D levels in patients with chronic disease [22]. ..."
Point 2: Line 195: Previous studies have pointed out that vitamin D deficiency is associated with many chronic diseases (autoimmune disease, cardiovascular disease, diabetes, obesity, mental disease and malignant tumor, etc.) except osteoporosis, caries, rickets and other vitamin D deficiency diseases in children……  “Except” is wrong. It should be ‘including”.
Response 2:Thanks for pointing out our error; we have made correction according to the Reviewer’s comments.
Reviewer 2 Report
The article “Serum 25-Hydroxy Vitamin D Levels in Children with Acute 2 Respiratory Infections Caused by Respiratory Virus or Atypical 3 Pathogen Infection” by Kuang et al. describes the levels of vitamin D in hospitalized children during viral infections. Thank you for giving me the opportunity to review it. The article is quite well written and easy to follow. The subject of vitamin D deficiency and viral infections is known but up to now, a relationship between various viral infections and vitamin D levels has not been described.
Nevertheless, I could have some questions and remarks to the authors.
1) Check carefully statistics and the way you present it. Statically insignificant differences are when p>0.05 not p>0.5. Check if it is not a simple typing error.
2) Were there no ICU admissions due to influenza B? Add some information or comment about it in Table 3.
3) In line 186, what about supplementation of any kind as a source of vitamin D? Please comment on it.
4) Was vitamin D supplemented during treatment?
5) Could the authors comment/explain why vitamin D levels rose after recovery?
6) Were the children on more or less the same diet (had the diet no direct influence on the fact of getting ill?)?
7) Could the authors comment why vitamin D levels were different in children with various infections? Could there be a real biological/medical cause other than the one related to statistics?
8) Please check the English language, sometimes there are small grammar mistakes
9) Add funding at the end of the text
10) An informed consent was obtained rather from the caregivers than from the patients themselves (both in the text and at the end of the article)
Reviewer 3 Report
1) Please be consistent when reporting numbers, e.g., in lines 54 and 55 "seventeen patients" and "two hundred and ninety-five hospitalized children" respectively while in lines 51 and 55 "948 children" and "636 healthy children" respectively.
2) Table 1 is difficult to read. Please add the appropriate statistic for each characteristic. What do you report for Age? Is it mean and 1st and 3rd quartiles in the parenthesis? For 25(OH)D levels you report mean +/- SD? why don't you have the SD for Age? Create a separate column for the Co-Infection group and add another p-value for the comparison between "Co-infection" and "Healthy" or Control group.
3) There should be only one p-value for the comparison in vitamin D status between ARIs and Healthy group (not 4 p-values).
4) Lines 123-124: "Our analysis showed that the prevalence of VDD and SVDD were higher in ARIs than that in healthy group (X2=94.66, p<0.0001)". The p-value shows that the difference is significant for all proportions not only for VDD and SVDD groups.
5) Lines 125-126: "Therefore, we suggest that children who have serum 25(OH)D levels below 50 nmol/L(VDD and SVDD status) are at increased risk of ARIs". There is also higher proportion of healthy patients in VDI group compared to ARIs. How do you explain this?
6) The pie chart (figure 1a) is not clear enough. It should show the percentages of HAdVs, PIV, EV etc.
7) The title of section 3.3 (lines 127-128) is confusing. What do you mean by "unremarkable"?
8) Figure 1 title is too long. Why don't you explain all this information in the text?
9) Table 3: What are the numbers in the parenthesis?
10) The authors found that "the prevalence of VDD and SVDD in children with ARIs was 226 significant higher than that in healthy children (59.66% vs. 31.13%)". This alone however is not enough to conclude that serum 25(OH)D levels below 50 nmol/L(VDD and SVDD status) are at increased risk of ARIs. You need to do a logistic regression analysis with ARIs vs Control group as dependent variable and serum 25(OH)D levels, age, sex as covariates to find whether the risk ratio for ARIs is statistically significant.
Round 2
Reviewer 3 Report
Point 4: I still don't understand what kind of multiple comparison tests you did for the vitamin D status. On the letter you write "The p-value shows that the difference is less significant for VDS and VDI groups (X2=8.008, p=0.05) rather than VDD and SVDD groups." . However, in the manuscript you wrote p-value = 0.005. It's better to delete the sentence "Additionlly, the proportion of healthy in VDS and VDI was higher compared to ARIs group, the p value showed that the difference was less significant for VDS and VDI (X2=8.008, p=0.005) rather than for VDD and SVDD in ARIs groups" because it is confusing.
You need to spend more time explaining the results of the logistic regression. Both in the results section as well as in the discussion
Author Response
Response to Reviewer 3 Comments
Point 1:
I still don't understand what kind of multiple comparison tests you did for the vitamin D status. On the letter you write "The p-value shows that the difference is less significant for VDS and VDI groups (X2=8.008, p=0.05) rather than VDD and SVDD groups." . However, in the manuscript you wrote p-value = 0.005. It's better to delete the sentence "Additionally, the proportion of healthy in VDS and VDI was higher compared to ARIs group, the p value showed that the difference was less significant for VDS and VDI (X2=8.008, p=0.005) rather than for VDD and SVDD in ARIs groups" because it is confusing.
You need to spend more time explaining the results of the logistic regression. Both in the results section as well as in the discussion.
Response 1:
Thanks again to the reviewer for the reasonable, scientific and patient revision suggestions.
a) We studied it carefully and found that our revision of the multiple comparison analysis was therefore not applicable here. We have removed the sentence that caused confusion and rewritten it as following: “…It is also shown that there is a higher percentage of VDS and VDI populations in the healthy group than in the ARIs group. In contrast to that, ARIs group has higher rates of VDD and SVDD than healthy group. …There are significant differences in vitamin D status between healthy and ARIs groups based on our analysis (X2=94.66, p<0.0001).”
b) As for the results and discussion of regression analysis, we also made some revisions as following:
Results section: “… The results showed that female children (OR=1.329, 95%CI: 1.007-1.755, P=0.045), >6 years old (OR=2.587, 95%CI: 1.946-3.438, P<0.0001), vitamin D status of VDD and SVDD (OR=3.264, 95%CI: 2.450-4.349, P<0.0001) was the risk ratio for ARIs, which was statistically significant. …”.
Discussion section: “Vitamin D deficiency has been reported as an increased risk of acute lower respiratory infections in children [39]. In our study, we found that female children, >6 years old or vitamin D levels <50 nmol/L may be risk factors for children with ARIs (Table 5). We suggest that children older than 6 years of age or who are female may be at greater risk of acute respiratory infections due to less exposure to sunlight during outdoor activities. This result is slightly different from previous reports that preschool girls with low vitamin D levels are more likely to be infected with the respiratory virus [42]; it may be attributed to the difference in geographical location and lifestyles in different countries. ”